# Antibody Response to Pertussis Vaccination in Pregnant and Non-Pregnant Women—The Role of Sex Hormones

**DOI:** 10.3390/vaccines9060637

**Published:** 2021-06-10

**Authors:** Victoria Peer, Khitam Muhsen, Moshe Betser, Manfred S Green

**Affiliations:** 1School of Public Health, University of Haifa, Abba Khoushy 199, Mount Carmel, Haifa 3498838, Israel; manfred.s.green@gmail.com; 2Department of Epidemiology and Preventive Medicine, School of Public Health, Sackler Faculty of Medicine, Tel Aviv University Ramat Aviv, Ramat Aviv, Tel Aviv 6139001, Israel; kmuhsen@tauex.tau.ac.il; 3Department of Obstetrics and Gynecology, The Yitzhak Shamir Medical Center (Formerly Assaf Harofeh Medical Center), Zerifin 70300, Israel; betserm@shamir.gov.il

**Keywords:** pertussis, vaccination, estrogen, progesterone, pregnancy, hormones

## Abstract

Pertussis containing vaccine is recommended for pregnant women to protect neonates prior to being fully immunized against the disease. The immune response during pregnancy may be impacted by changes in the hormonal status. The aim of this study was to evaluate the immune response to pertussis immunization in pregnancy and to assess the role of sex hormones. In a cross-sectional study, blood samples were drawn from 174 pregnant and 74 non-pregnant women 45–60 days following immunization. Anti-pertussis toxin (Anti-PT) IgG antibody levels, estrogen, and progestogen concentrations were compared between the two groups. Multiple logistic regression analysis was used to examine the association between serum antibody and sex hormone concentrations in each group, controlling for age, body mass index (BMI), and smoking status. The geometric mean concentration (GMC) of anti-PT IgG antibody was significantly higher in non-pregnant women compared with pregnant women (median of 2.09 and 1.86, interquartile range = 2.36–1.8 and 2.11–1.16 respectively, *p* < 0.0001). Among pregnant women, the anti-PT IgG antibody GMC was negatively associated with both progesterone (odds ratio = 0.300, 95% CI = 0.116, 0.772, *p* = 0.013) and estrogen (odds ratio = 0.071, 95% CI = 0.017, 0.292, *p* < 0.0001), after controlling for age, BMI, and smoking. Pregnancy was associated with lower anti-PT IgG antibody levels (odds ratio = 0.413, 95% CI = −0.190, 0.899, *p* = 0.026). This appears to be at least partially explained by the higher levels of hormones during pregnancy. These findings demonstrate the important role of sex hormones in the response to pertussis vaccine during pregnancy and can help to evaluate the optimum vaccination schedule.

## 1. Introduction

Pertussis is a highly contagious, vaccine-preventable disease caused by *Bordetella pertussis* (B. pertussis). The whole cell vaccine against pertussis, combined with diphtheria and tetanus toxoids, has long been part of the routine immunization schedule for infants and young children. Since the early 1990s, an acellular vaccine has been introduced in many countries [1]. Although the acellular vaccine is highly effective, it is less effective in preventing the spread of the disease [2]. Pertussis vaccination was included in the National Immunization Program in Israel in 1957 and since 2005, has been given together with tetanus and diphtheria toxoids at two, four, six, and 12 months, at 7–8 years, and 13–14 years [3]. Despite a high vaccination infant coverage rate (>93%) in Israel, there is still a considerable circulation of B. pertussis, particularly among 15–19 year-olds as well as in the older age cohort (>60 years) [4]. Following many years of declining incidence rates, over the past few years there has been a resurgence of pertussis [5,6].

Pertussis maternal antibodies cross the placenta and provide the newborn with protection against pertussis in early life [7]. In 2012, in an effort to ensure high levels of maternal antibodies, the United States Advisory Committee on Immunization Practices (ACIP) [7] recommended a dose of pertussis-containing vaccine for pregnant women between 27 and 36 weeks gestation at every pregnancy. Maternal pertussis vaccination during pregnancy became part of the National Immunization Program in Israel in 2015 [8]. Attitudes towards pertussis vaccine, beliefs about safety, effectiveness, and the timing of communication about vaccination are important determinants of vaccine acceptance during pregnancy [9].

Pregnancy is associated with substantial changes in concentrations of sex hormones including estradiol, estriol, progesterone and prolactin and characterized by a period of immune quiescence [10,11]. It is a unique system of pro- and anti-inflammation processes. Changes in cellular and molecular processes facilitate maternal immune adaptations that are only partly understood [10,11]. The immune environment reflected in peripheral blood adapts to sustain immune tolerance [12]. There is evidence that the hormonal changes in pregnancy may affect the immune response to vaccines [11,13]. Findings on the effect of pregnancy on pertussis antibody levels following vaccination have been variable [14,15,16,17,18]. In one study, post-vaccination titers against pertussis toxin and filamentous hemagglutinin were significantly higher in non-pregnant versus pregnant women [14]. Other studies found that antibody responses to Tdap vaccine in pregnant women were not different from those of non-pregnant women and they increase with the same extent [15,16]. For influenza vaccines given in pregnancy, the antibody response was similar in pregnant and non-pregnant women [17,18].

Assessing the effect of pregnancy on the immune response to the pertussis vaccine is important for determining the timing of vaccination for optimal protection of the infant prior to active immunization. The goal of this study was to examine the immune response to pertussis vaccine in pregnancy and the association of estrogen and progesterone levels with the immune response.

## 2. Methods

### 2.1. Study Design

A cross-sectional study was conducted in the Shamir (Asaf Harofe, Be’er Ya’akov, Israel) Medical Centre between 2017 and 2018.

### 2.2. Study Population

Healthy pregnant women 18–45 years-old were recruited from the obstetric departments at a general hospital. Inclusion criteria were first pregnancy with a single foetus, no underlying chronic or pregnancy related medical conditions and vaccinated during pregnancy with pertussis-containing vaccine. The comparison group, comprising non-pregnant women aged 18–45 years, was recruited from the occupational clinic at the hospital. In accordance with the Israeli Ministry of Health regulations, every new employee in a medical institution must be vaccinated against pertussis. The candidates were recruited in the occupational clinic, which is responsible for administering and documenting the vaccines for all employees in the medical centre. Inclusion criteria were healthy women who do not take birth control pills, with no underlying comorbidities, and who were vaccinated as part of their employment in the hospital. All participants received an explanation on the background and the purpose of the study, signed an informed consent form to participate in the study, provided their details, and filled out the questionnaires. Medical records were checked to verify the pertussis vaccine administration.

Pregnant and non-pregnant women who had previously received the pertussis containing vaccine or who had documented pertussis within the previous five years were excluded. In order to rule out vaccination in recent years, only pregnant women with a first delivery child were included in the study. In addition, every woman was asked about getting vaccinated for any reason. Pregnant women were asked about weight in pregnancy, so the BMI value of pregnant women is a derivative of their weight at the time of receiving the vaccine. Parameters such as smoking status, height and weight were self-reported. All participants were vaccinated between years 2017–2018. A total of 74 non-pregnant women (control group) and 174 pregnant women immediately prior to labour were recruited. Institutional Review Board Statement: The institutional review board at Shamir (Asaf Harofe) Medical Center, Israel and ethics committee at Haifa University, Israel approved the study protocol. Approval numbers are 0290-17-ASF and 0181-17-ASF.

### 2.3. Blood Sampling

Blood samples were drawn 45–60 days after the immunization and centrifuged within an hour of collection. Sera samples were stored at −70 °C until tested for anti-pertussis toxin (anti-PT) IgG antibody, estrogen, and progesterone levels.

### 2.4. Questionnaires

Questionnaires were administered to women in both groups contained questions about their age, weight, height, time of the vaccination during pregnancy, chronic diseases, smoking status and use of medication on a regular basis. Body mass index (BMI) was calculated as weight (in kilograms)/height (in meters)^2^.

### 2.5. Study Vaccine

In both groups, the licensed pertussis vaccine (Tdap, Boostrix^®^, GlaxoSmithKline, Middlesex, UK) was administered as a 0.5-mL intramuscular injection containing 5 Lf of tetanus toxoid, 2.5 Lf of diphtheria toxoid, 8 µg of inactivated PT, 8 µg of FHA (Filamentous Hemagglutinin), and 2.5 µg of pertactin.

### 2.6. Antibody Assays

The enzyme-linked immunosorbent assay (ELISA) was used for quantitative determination of anti-PT IgG antibody in serum (expressed in International Units per milliliter, IU/mL). The presence of pertussis IgG antibodies was detected by using a commercially available ELISA kit (EUROIMMUN, Lübeck, Germany). The lower detection limit of the quantitation for pertussis toxin IgG was 0.2 IU/mL.

### 2.7. 17β-Estradiol and Progesterone Quantitative Determination

Electrochemiluminescence immunoassay (ECLIA) was performed for 17β-estradiol and progesterone quantitative determination. Further, 17β-estradiol and progesterone values below the limit of detection are reported as <5 pg/mL and <0.05 ng/mL, respectively.

### 2.8. Statistical Analyses

In the current study we tested pre-defined hypotheses of association between pregnancy status, estrogen and progesterone levels with anti-PT IgG antibody levels. For the association with the hormones, we planned stratified analyses separately for pregnant and non-pregnant women. The levels of estrogen and progesterone are highly correlated. Thus, to avoid multicollinearity, these variables were analyzed in separate models. Other variables such as age, body mass index, and smoking were included in all models as potential confounders.

The characteristics of the groups were described using means, standard deviations (SD), medians, minimum and maximum and the inter-quartile range for continuous variables. Percentages were computed for categorical variable. The main independent variable was pregnancy status, additional variables were selected as covariates (age, BMI, smoking status). The main outcome variable was the anti-PT IgG antibody concentration.

The scatter plots were used to visually display the relationship between the variables (anti-PT IgG antibody, estrogen, progesterone, age, BMI and smoking status) in each group of pregnant and non-pregnant women. The data on antibody, estrogen, and progesterone concentrations were transformed to natural logs in order to calculate the Geometric Mean Concentration (GMC). Continuous variables were tested for normality using the Shapiro-Wilk test.

The log-transformed antibody, estrogen and progesterone concentrations were compared between the pregnant and non-pregnant women groups using the Mann-Whitney test. Correlations between independent variables were tested using Pearson’s correlation. Multicollinearity was assessed using variance inflation factor (VIF). According to the manufacturer protocol, the anti-PT IgG antibody level <40 IU/mL was defined as negative, and antibody level ≥40 IU/mL was assumed to be positive. Concentration of 40 IU/mL was defined as anti-PT IgG antibodies cut-off value. Multiple logistic regression analysis was used to compute odds ratios (OR) for the association of the dichotomous antibody level with pregnancy status (pregnant or non-pregnant) after adjusting for age, BMI, and smoking status. Multiple logistic regression analysis was also used to examine the association between anti-PT IgG antibody following immunization and serum hormones (estrogen and progesterone) in each group of pregnant and non-pregnant women, controlling for age, BMI and smoking status. All analyses were performed using SPSS-27 software, IBM, Armonk, NY, USA.

## 3. Results

The demographic characteristics of the groups are shown in Table 1.

The mean age of control group was 24.5 years (SD = 8.5) and the mean age of women at birth was 26.0 (SD = 3.3).

The scatter plots for the visualization of the results are shown in Figure 1 and Figure 2.

Scatter plots of the levels of anti-PT IgG antibody (IgG, IU/mL), estrogen (pg/mL), and progesterone (ng/mL) show no correlation in non-pregnant women. In pregnant women, the antibody level tends to decrease as the estrogen and progesterone levels increase.

Scatter plots (Figure 2) of level of anti-PT IgG antibody (IgG, IU/mL), age (years) and BMI show no correlation in pregnant women. In non-pregnant women, the antibody level tends to decrease as the age increases.

### 3.1. Comparison of Anti-PT IgG Antibody Levels between Pregnant and Non-Pregnant Women

The post-vaccination anti-PT IgG antibody levels (GMC) were compared between the pregnant and non-pregnant groups (Figure 3). In both groups the anti-PT IgG antibody levels were above the detection limit of 0.2 IU/mL. Pregnant women had significantly lower GMC compared to non-pregnant women, with a median of 1.86 and 2.09, and interquartile range (IQR) = 2.11–1.16 and 2.36–1.8 respectively, *p* < 0.0001.

### 3.2. Multiple Logistic Regression Analysis—Pregnancy Status (Pregnant vs. Non-Pregnant Women) Assosiation with Anti-PT IgG Antibody Levels, Controlling for Other Variables

The results of the multiple logistic regression analysis of the association of pregnancy status with anti-PT IgG antibody levels, after controlling for age, BMI, and smoking, are shown in Table 2. A significant negative association was found between pregnancy status and anti-PT IgG antibody levels (OR = 0.413, 95% CI 0.190, 0.899, *p* = 0.026).

### 3.3. Estrogen and Progesterone Levels in Each Group of Pregnant and Non-Pregnant Women

Compared to non-pregnant women, pregnant women exhibited significantly higher serum estrogen and progesterone levels. The Mann–Whitney test revealed differences in progesterone (GMC) concentration between pregnant and non-pregnant women, median = 2.86 (IQR = 2.99–2.64) and 0.39 (IQR = 1.38–(−0.11) respectively, *p* < 0.0001. The median estrogen GMC in pregnant and non-pregnant women was 4.86 (IQR = 4.99–4.63) and 2.47 (IQR = 2.67–2.07), respectively, *p* < 0.0001.

### 3.4. Multiple Logistic Regression Analyses of the Association between Anti-PT IgG Antibody Levels and Sex Hormones

#### Association between Anti-PT IgG Antibody Levels and Estrogen

Multiple logistic regression analysis of the association between anti-PT IgG antibody levels (cut-off of 40 IU/mL) and serum estrogen (GMC) in non-pregnant and pregnant women controlled for age, BMI, and smoking status is shown in Table 3.

In non-pregnant women group, significant negative association was found between age and levels of anti-PT IgG antibody (OR = 0.911, 95% CI 0.848, 0.978, *p* = 0.010). Among pregnant women, serum estrogen levels were negatively related to the levels of anti-PT IgG antibody (OR = 0.071, 95% CI 0.017, 0.292, *p* < 0.0001) (Table 3).

### 3.5. Association between Anti-PT IgG Antibody Levels and Progesterone

Multiple logistic regression analysis of the association between anti-PT IgG antibody levels (cut-off of 40 IU/mL) and serum progesterone (GMC) in non-pregnant and pregnant women, controlled for age, BMI, and smoking status, is shown in Table 4.

The multiple logistic regression model for the effect of individual variables (progesterone, age, BMI, and smoking) on anti-PT IgG antibody levels in non-pregnant women showed that increased age was significantly negatively associated with antibody level (OR = 0.912, 95% CI 0.850,0.979, *p* = 0.011), but no significant association was found with progesterone level. In pregnant women, the logistic regression model (controlled by age, BMI and smoking status) demonstrated a negative association between progesterone level and anti-PT IgG antibody levels (OR = 0.300, 95% CI 0.116, 0.772, *p* = 0.013) (Table 4).

In the cohort of pregnant women, we found a statistically significant positive correlation between progesterone and estrogen (Pearson correlation = 0.89, *p* < 0.0001). We chose to perform another test of multicollinearity between the variables using the variance inflation factor [19]. According to this test, the correlations between VIF values (VIF > 5 for estrogen and progesterone, *p* < 0.0001 and *p* = 0.006 respectively) were high, meaning it is difficult to separate the independent effect of estrogen and progesterone in the model. As a result, we decided not to include estrogen and progesterone in the same model of logistic regression.

## 4. Discussion

We compared anti-PT IgG antibody concentration after the vaccination in pregnant and non-pregnant women and examined the association of sex hormones with the antibody levels in each group separately. Our findings revealed that, compared with non-pregnant women, pregnant women who were vaccinated against pertussis during the third trimester in pregnancy developed significantly lower anti-PT IgG antibody levels. Among pregnant women, the level of pertussis IgG antibodies was negatively associated with both progesterone and estrogen.

### 4.1. There Are Several Strengths and Limitations of the Study

There could be some selection bias, since all participants were selected from one hospital, which generally represents one geographical area in the country. However, there does not seem to be any reason why the results should not be generalizable to other populations. As regards information bias, data on height, weight, smoking status, and comorbidities were self-reported. There may be incompleteness of the reports of the comorbidities. There is no reason to suspect that the extent of possible inaccuracies would differ substantially between the groups. The information about the last vaccination by pertussis containing vaccine is based on medical records and should be reliable in both groups. In pregnant women, hormone levels were not measured at the time of vaccination, but 45–60 days after. Since serum estrogen and progesterone levels remain high until the end of the third trimester of pregnancy [20,21,22,23] and change only after delivery [24,25], the hormone levels measured immediately before delivery should be representative of the hormone levels at the time of immunization. As regards the anti-PT immunological memory in the study population, due to previous exposure or immunization, there does not appear to be any reason to believe that the extent of immunological memory in pregnant and non-pregnant groups will be different.

### 4.2. Comparison with Other Studies

Concentrations of estrogens and progesterone are considerably increased over the course of pregnancy, with highest levels achieved during the third trimester [7] when pregnant women are vaccinated. Our results are in agreement with the study of pregnant and non-pregnant women vaccinated with Tdap, where anti-PT IgG antibody titers were determined 28 days post-vaccination [14]. They found that antibody levels against pertussis toxin and filamentous hemagglutinin were significantly higher in non-pregnant women.

In two studies [15,16], there was no evidence of differences in the antibody responses to Tdap vaccine in pregnant and non-pregnant women. Pertussis IgG antibody levels increased significantly and to the same extent after vaccination in pregnant and non-pregnant women [15,16].

In both studies [15,16], the small number of participants were enrolled. Prior vaccination information about receiving the vaccine was obtained mainly through self-reporting which is a possible source of information bias [16]. It is possible that both studies were underpowered to detect any differences in the antibody responses between pregnant and non- pregnant females.

### 4.3. Possible Mechanisms

We believe that differences in pertussis IgG antibody levels that we observed between pregnant and non-pregnant women may be the result of immune and other profound alterations during pregnancy that cause unique and diverse hormonal, metabolic and physiological changes. According to our findings, a high level of reproductive hormones correlate with low vaccine response. Since pregnancy is characterized by high levels of estrogen and progesterone [10], the activation of immune responses to vaccines may be altered by pregnancy [26] with increased susceptibility to different pathogens [27]. This suggests that hormones or the hormone-associated physiological status contributes to the immune response to pertussis vaccination.

High estradiol concentrations, usually encountered during pregnancy, lead to CD4+ type 2 helper T-cell (Th2) responses [28,29]. This mechanism, responsible for successful pregnancy maintenance, is related to a switch from the T helper 1 profile to the T helper 2 profile. Th17 cells (part of pertussis post vaccination response) involved in successful pregnancy, and Th1/Th2/Th17 paradigm is a part of complicated immune cells interactions [30] and pertussis vaccine mediated immune responses. Progesterone has suppressive role in general, and in pregnancy it influences the balance between Th1 and Th2 immune responses [31,32] and participates (in animal model) in Th17 expression inhibition in a dose-dependent manner [33].

The complete immunological mechanisms of the protection provided by acellular pertussis (aP) vaccines are not fully described. It is possible that antibody-mediated and Th1/Th17 immune responses are important [34,35]. Acellular pertussis vaccines, such as Boostrix, promote the immune responses to a mixed Th2/Th1 profile, and enhance antibody production after the vaccination [36]. Complex T- and B- cell immune responses to pertussis vaccine have been demonstrated [37]. Protective immune responses are depended not only on antibodies but also on CD4+ T cell and B cells. B and T cell cross-talk is required for the optimal maintenance of functional B as well as T-cell memory to pathogen [38]. B cells contribute to protective immunity to B. pertussis in mice by activating CD4+ cells or by producing chemokines and cytokines [39]. Memory B cells are crucial for the pertussis IgG antibodies production [40]. CD4+ T cells secreting both IL-9 and IL-17 have been shown to be associated with pertussis-specific responses after the vaccination. IFN-β is regulator of both IL-9 and IL-17 [41].

We believe that the association that we found between progesterone, estrogen and pertussis IgG antibody levels could be explained, at least in part, by the effect of sex hormones on B cells. The characteristics of peripheral B cell compartment differ significantly between pregnant and non-pregnant women [42] and the inverse association found in this study could be explained by the immunosuppressive activity of estrogen levels in pregnancy and the activity of progesterone.

The number and activity of B lymphocyte precursors in the bone marrow have been demonstrated in normal pregnant mice, suggesting that B lymphopoiesis is sensitive to negative regulation by sex steroids [43,44]. The inhibitory effects of elevated estrogens suppress adult B-lymphopoiesis during pregnancy [45]. In animal models, progesterone represses the differentiation and maturity of B cells [46]. The interplay between hormone levels, the activity of immune cells post vaccination, and the immunosuppression in pregnancy may explain the lower anti-PT IgG antibody levels.

## 5. Conclusions

Pregnancy is a unique period that combines hormonal interplay and complex systems of pro- and anti-inflammation immune responses. The impact of sex hormones on the response to pertussis vaccine should be taken into account when evaluating the immune response to vaccination in pregnancy. Clinical and epidemiological studies should include the influence of the reproductive and hormonal status on vaccine-induced immunity.

## Figures and Tables

**Figure 1 vaccines-09-00637-f001:**
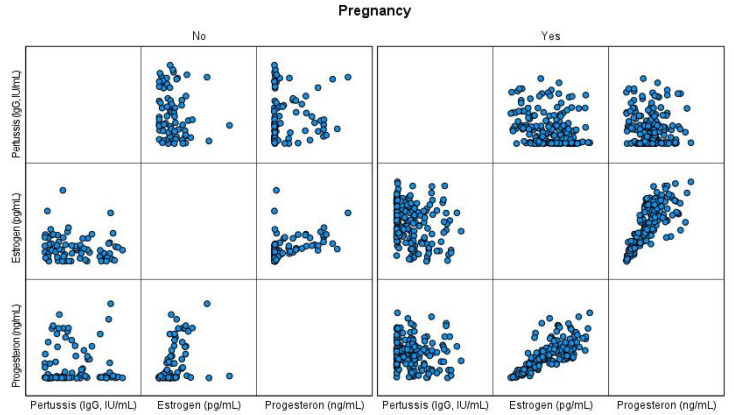
Scatter plot for anti-PT IgG antibody, estrogen and progesterone levels for pregnant and non-pregnant women.

**Figure 2 vaccines-09-00637-f002:**
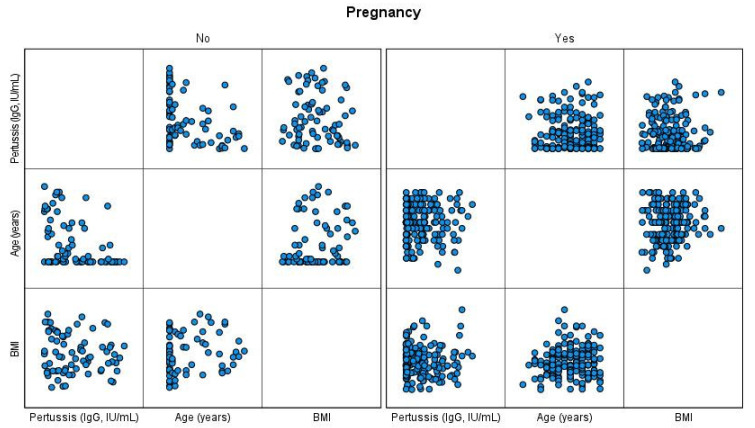
Scatter plots for anti-PT IgG antibody, age and BMI in pregnant and non-pregnant women.

**Figure 3 vaccines-09-00637-f003:**
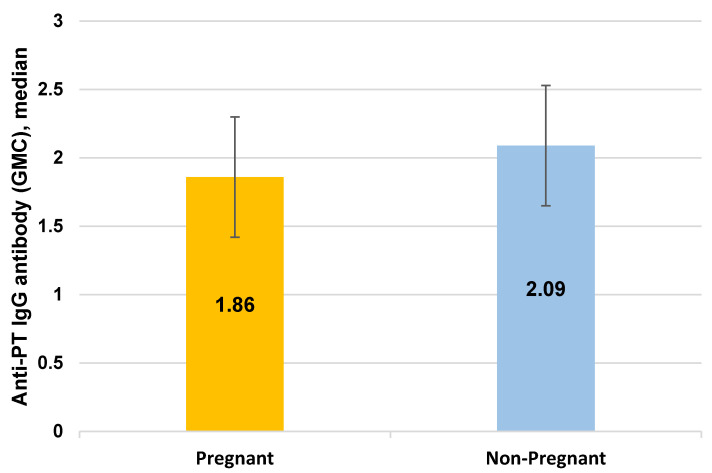
Anti-PT IgG antibody levels (GMC) in serum from pregnant (*n* = 174) and non-pregnant (*n* = 74) women. GMC = Geometric Mean Concentration.

**Table 1 vaccines-09-00637-t001:** Demographic and clinical characteristics of non-pregnant and pregnant women.

Characteristics of Study Participants	Non Pregnant Women (*n* = 74)	Pregnant Women (*n* = 174)
Ethnicity, *n* (%)		
Jews	67 (90.5%)	148 (85.0%)
Arabs	7 (9.5%)	26 (15.0%)
Age (years)		
Mean (SD)	24.5 (8.5)	26.0 (3.3)
Median (min, max)	19.5 (18.0, 45.0)	26.0 (18.0, 31.0)
Height (cm)		
Mean (SD)	162.5 (7.2)	163.0 (6.8)
Median (min, max)	161.0 (147.0, 182.0)	162.0 (148.0, 184.0)
Weight (kg)		
Mean (SD)	58.5 (9.2)	75.3 (13.3)
Median (min, max)	57.0 (42.0, 8)	74.0 (50.0, 133.0)
BMI kg/m^2^		
Mean (SD)	22.1 (3.2)	28.4 (4.4)
Median (min, max)	21.7 (16.4, 28.7)	27.8 (20.0, 44.0)
Smoking status, *n* (%)		
Yes	11 (14.9)	15 (9.0)
No	63 (85.1)	159 (91.0)

SD = Standard Deviation; BMI = Body Mass Index; *n* = Number of participants.

**Table 2 vaccines-09-00637-t002:** Multiple logistic regression analysis of the association between levels of anti-PT IgG antibody (GMC) with pregnancy status, age, BMI, and smoking.

Variables in the Model	β Coefficient	Odds Ratio	95% CI	*p*
Pregnancy (Yes vs. No)	−0.885	0.413	0.190, 0.899	0.026
Age, years	−0.044	0.957	0.906, 1.011	0.119
BMI, kg/m^2^	−0.024	0.976	0.914, 1.043	0.470
Smoking, (Yes vs. No)	0.034	1.034	0.495, 2.161	0.928

GMC = Geometric Mean Concentration; *n* of non-pregnant women = 74; *n* of pregnant women = 174; BMI = Body Mass Index.

**Table 3 vaccines-09-00637-t003:** Multiple logistic regression analysis for the association between anti-PT IgG antibody levels with estrogen, age, BMI, smoking status in non-pregnant and pregnant women.

Variables in the Model	95% CI	Odds Ratio	β Coefficient	*p*
Non-pregnant (n = 74)				
Estrogen (GMC)	−0.663	0.515	0.134, 1.975	0.333
Age, years	−0.094	0.911	0.848, 0.978	0.010
BMI, kg/m^2^	−0.134	0.875	0.713, 1.072	0.197
Smoking, (Yes vs. No)	−1.472	0.229	0.023, 2.251	0.206
Pregnant (n = 174)				
Estrogen (GMC)	−2.641	0.071	0.017, 0.292	<0.0001
Age, years	0.053	1.054	0.946, 1.175	0.340
BMI, kg/m^2^	−0.012	0.988	0.916, 1.066	0.758
Smoking, (Yes vs. No)	0.587	1.799	0.733, 4.416	0.200

GMC = Geometric Mean Concentration; BMI = Body Mass Index.

**Table 4 vaccines-09-00637-t004:** Multiple logistic regression analysis for the association between anti-PT IgG antibody levels with progesterone, age, BMI, smoking status in non-pregnant and pregnant women.

Variables in the Model	95% CI	Odds Ratio	β Coefficient	*p*
Non-pregnant (n = 74)				
Progesterone (GMC)	0.489	1.631	0.713, 3.733	0.246
Age, years	−0.092	0.912	0.850, 0.979	0.011
BMI, kg/m^2^	−0.123	0.885	0.723, 1.083	0.235
Smoking, (Yes vs. No)	−1.560	0.210	0.021, 2.140	0.188
Pregnant (n = 174)				
Progesterone (GMC)	−1.205	0.300	0.116, 0.772	0.013
Age, years	0.066	1.068	0.962, 1.186	0.220
BMI, kg/m^2^	−0.011	0.989	0.920, 1.063	0.760
Smoking, (Yes vs. No)	0.458	1.581	0.673, 3.713	0.294

GMC = Geometric Mean Concentration; BMI = Body Mass Index.

## Data Availability

The data that support the findings of this study are available on request from the corresponding author.

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
