# Peer review of "Antibody Response to Pertussis Vaccination in Pregnant and Non-Pregnant Women—The Role of Sex Hormones"

_vaccines, 2021, doi:10.3390/vaccines9060637_

Round 1

Reviewer 1 Report

This study makes an attempt to answer the question of why the levels of antibodies from the Tdap Vaccine are lower in pregnant women than in non-pregnant women.
A negative relationship is found between hormone levels (estrogens and progesterone) and antibody levels in pregnant women.
This may partly answer the question of why pregnant patients have a lower immune response.

My opinion is that this paper is suitable for publication with minor changes:

In line 63 you comment, at the end of the introduction: "Assessing the effect of pregnancy on the immune response to the pertussis vaccine is 63 important for determining the timing of vaccination for optimal protection of the infant 64 prior to active immunization."But later, in the discussion and conclusions, no comment is made on this point. I would welcome any comment on whether the current vaccination time is the most suitable.

Thak you

Author Response

Response:

We thank you for your useful comment and suggestion. 

The impact of sex hormones on the response to pertussis vaccine should be taken into account when evaluating the immune response to vaccination in pregnancy. Clinical and epidemiological studies should include the influence of the reproductive and hormonal status on vaccine-induced immunity. Non-pregnant and pregnant women should be considered as different groups when studying vaccine efficacy.

However, the current study didn't designed to recommend the most suitable vaccination time to provide the optimal protection for the infant

We have rewritten to make it clearer (lines 363-368)

Reviewer 2 Report

In a cross-sectional study, blood samples were drawn from 174 pregnant (3rd trimenon) and 74 non-pregnant women 45-60 days following immunization with Pertussis containing vaccine (Tdap; Boostrix). Anti-Bordetella pertussis toxin IgG antibody levels (anti-PT IgG), estrogen and progestogen concentrations were compared between the two groups using the Mann-Whitney test.

In pregnant females, anti-PT IgG tended to decrease as the estrogen and progesterone levels increased, while in non-pregnant females, anti-PT IgG tended to decrease with increasing age. Pregnant women had a significantly lower post-vaccination anti-PT IgG level (geometric mean concentration; GMC) compared to non-pregnant women. Multiple logistic regression analysis revealed a significant negative association between pregnancy status and anti-PT IgG level, after controlling for age, BMI and smoking.

The authors thus assume that the differences in PT IgG levels observed between pregnant and non-pregnant females may be the result of "immune and other profound alterations that cause unique and diverse hormonal, metabolic and physiological changes" that occur during pregnancy. Especially a "high, pregnancy level of hormones has a suppressing effect on vaccine response."

Major points that need to be clarified by the authors:

Methods:

Some information on previous exposure to pertussis in the study population should be given. Any information on previous exposure available (e.g., previous disease or vaccination against pertussis before pregnancy)? Are there any data on pertussis seroprevalence in the population studied?

Discussion:

The role of pre-existing pertussis-specific immunity as an important confounder of the authors' findings should be addressed. Is there any hint of anti-PT immunological memory in the study population? Could there be differences in previous exposure to pertussis (either disease or vaccination)?

The discussion should be more concise and focus on the immune response to pertussis: e.g. line 238-245: the study does not adress differences between immune responses between different pathogens or vaccines such as pertussis toxin, HPV 16/18, Covid-19, or 23-valent pneumococcal vaccine - so why should it be discussed? Line 246-256: only women were studied, so why discuss sex differences in immune responses (without giving any precise hint to physiological mechanisms that may be relevant for interpretation of study results)?

Most importantly, the authors should describe the differences between their study and previous studies on that topic in more detail (line 228-237), given the fact that the cited studies either did not include non-pregnant women or did explicitly assess immune responses before and after vaccination.

On the other side, a more comprehensive description of anti-pertussis immune responses including information on T- and B-cell memory as well as functional aspects (e.g., antibody avidity) is lacking.

Minor points:

It remains unclear how the authors did correct their statistical analysis for multiple comparison.

Author Response

Reviewer 2

We thank you for the detailed review and useful comments and suggestion.

Methods:

1.Some information on previous exposure to pertussis in the study population should be given. Any information on previous exposure available (e.g., previous disease or vaccination against pertussis before pregnancy)?

Response:

Medical records were checked to verify the pertussis vaccine administration. Pregnant and non-pregnant women who had previously received the pertussis containing vaccine or who got sick within the prior 5 years were excluded. In order to rule out vaccination in recent years only pregnant women with the first child participated in the study. In addition, every woman was asked about getting vaccinated for any reason. No booster doses were given 5 years before of serum sampling.

This is now clarified in the revised text (Methods, lines 97-102)

Are there any data on pertussis seroprevalence in the population studied?

Response:

Despite a high vaccination infant coverage rate (>93%) in Israel, there is still a considerable circulation of B. pertussis, particularly among 15–19 year-old subjects as well as in the older age cohort (>60 years) .

This is now clarified in the revised text (Introduction, lines 41-43)

Discussion:

The role of pre-existing pertussis-specific immunity as an important confounder of the authors' findings should be addressed. Is there any hint of anti-PT immunological memory in the study population? Could there be differences in previous exposure to pertussis (either disease or vaccination)?

Response:

Medical records were checked to verify the pertussis vaccine administration. Pregnant and non-pregnant women who had previously received the pertussis containing vaccine or who got sick within the prior 5 years were excluded. In order to rule out vaccination in recent years only pregnant women with the first child participated in the study. In addition, every woman was asked about getting vaccinated for any reason. No booster doses were given 5 years before of serum sampling.

This clarified in the revised text (Methods, lines 97-102).

As regards the anti-PT immunological memory in the study population, due to previous exposure or immunization, there does not appear to be any reason to believe that the immunological memory in pregnant and non-pregnant groups will be different.

Thus, there should be no difference between the two groups.

This clarified in the revised text (Discussion, lines 257-259).

The discussion should be more concise and focus on the immune response to pertussis: e.g. line 238-245: the study does not address differences between immune responses between different pathogens or vaccines such as pertussis toxin, HPV 16/18, Covid-19, or 23-valent pneumococcal vaccine - so why should it be discussed?

Response:

 The pathogens / vaccines mentioned are intended to emphasize the significance of age at the time of vaccination. The aim was to note that even when other vaccines are given to people at young ages (as our study participants), "age" is an important factor that influences vaccine responses.

We have rewritten the paragraph (Discussion, lines 283-291).

 Line 246-256: only women were studied, so why discuss sex differences in immune responses (without giving any precise hint to physiological mechanisms that may be relevant for interpretation of study results)?

Response:

We agree with you and have rewritten the paragraph as follows (Discussion, lines 296-311).

Host factors including age and pregnancy play important roles as modifiers of vaccines efficacy. The activation of immune responses to vaccines may be altered by pregnancy. Pregnant women are not immunosuppressed, but their immune system responses are shifted toward an anti-inflammatory phenotype . These facts can lead us towards the assumption that hormones or hormones associated different physiological status contribute in immune system response on vaccination.

Most importantly, the authors should describe the differences between their study and previous studies on that topic in more detail (line 228-237), given the fact that the cited studies either did not include non-pregnant women or did explicitly assess immune responses before and after vaccination.

Response:

Our results are in agreement with the study of pregnant and non-pregnant women vaccinated with Tdap, where Pertussis toxin IgG antibody titers were determined 28 days post-vaccination . They found that antibodies levels against pertussis toxin and filamentous hemagglutinin were significantly higher in non-pregnant women. In contrast, Munoz et al.  and Huygen et al.  found that the antibody responses to Tdap vaccine in pregnant women were not different from those of non-pregnant women and vaccination can increase vaccine specific IgG antibodies to the same extent in pregnant and in non-pregnant women. In the study of Munoz FM et al. 49 healthy pregnant women and 32 healthy non-pregnant women were enrolled. Immunogenicity assessments tested on blood samples from pregnant and non-pregnant women elicited immune responses similar to non-pregnant women. In a small sample, Huygen K et al. found that anti-pertussis toxin (PT) levels increased significantly and to the same extent after vaccination in pregnant and non-pregnant women. Proliferative and IFN-γ responses were in-creased to PT in non-pregnant women, whereas the stimulation of vaccine specific Th1 type cellular immune responses with vaccine was impaired during pregnancy.

We have rewritten the paragraph (Discussion, lines 266-281).

On the other side, a more comprehensive description of anti-pertussis immune responses including information on T- and B-cell memory as well as functional aspects (e.g., antibody avidity) is lacking.

Response:

Protective immune responses against B. pertussis are depended not only on antibodies but also on CD4+ T cell and B cells. B and T cell cross-talk is required for the optimal maintenance of functional B as well as T-cell memory to pathogen. In mice, Leef et al.  found that B cells contributed to protective immunity to B. pertussis by activating CD4+ cells or by producing chemokines and cytokines.  Memory B cells are crucial for the pertussis IgG antibodies production . CD4+ T cells secreting both IL-9 and IL-17 have been shown to be associated with pertussis-specific responses after the vaccination. IFN-β is regulator of both IL-9 and IL-17.

We have rewritten the paragraph (Discussion, lines 317-324).

Minor points:

It remains unclear how the authors did correct their statistical analysis for multiple comparison.

Response:

Due to the possibility of some multiple comparisons, the p-values should be interpreted with caution.

This is now clarified in the revised text (Methods, lines 152-153)

Reviewer 3 Report

The authors evaluate the immune response to pertussis immunization in pregnancy and assess the role of sex hormones. This is a very interesting topic. In general, this is a potentially stimulating article. I have some minor suggestions for your manuscript.

  1. Your introduction could better inform the reader about the context of the Israel Immunization Program. New issues and challenges to social-psychological factors associated with maternal pertussis vaccination acceptance and vaccination practices among health professionals could be included and discussed to increase the audience's interest.
  2. More detail concerning the data collection process is needed, namely to clarify the recruitment process, and eligibility criteria (never pregnant women in the control group?) ethic procedures, participation rate. Please describe in detail the setting, the procedures for data collection (who administered the questionnaire?), and the measures (indicate that weight and height were self-reported. How was BMI calculated for pregnant women? Why were only BMI and smoking selected to be independent variables? How was smoking status collected? Were there other socio-demographic characteristics besides age and ethnicity – which needs to be referred to in the methods section -, such as nationality and educational level collected?).
  3. Implications of the results for public policy should be enlightened. Future recommendations for research and action should be added.
  4. The English language (style and grammar) could be improved (some typos, consider replacing pregnant and non-pregnant female with pregnant and non-pregnant women).

Author Response

Reviewer 3

We are grateful for comprehensive review and important suggestions.

I have some minor suggestions for your manuscript.

  1. Your introduction could better inform the reader about the context of the Israel Immunization Program. New issues and challenges to social-psychological factors associated with maternal pertussis vaccination acceptance and vaccination practices among health professionals could be included and discussed to increase the audience's interest.

Response:

Pertussis vaccination was included in the National Immunization Program in Israel in 1957, and since 2005 DTaP is given at 2, 4, 6, and 12 months, at 7–8 years and 13–14 years. Despite a high vaccination infant coverage rate (>93%) in Israel, there is still a considerable circulation of B. pertussis, particularly among 15–19 year-old subjects as well as in the older age cohort (>60 years). Following many years of declining incidence rates, over the past few years there has been a resurgence of pertussis.

Maternal pertussis vaccination during pregnancy became part of the National Immunization Program in Israel in 2015 . Attitude towards pertussis vaccine, beliefs about safety, effectiveness and the timing of communication about vaccination can be essential determinants of vaccine acceptance during pregnancy.

In accordance with the work procedures of the Israeli Ministry of Health, every new employee hired to work in a medical institution must be vaccinated against pertussis. The candidates are recruited in the occupational clinic, which is responsible for administering and documenting the vaccines for all employees in the medical centre.   

We have rewritten the paragraph (Introduction, lines 39-45, 51-55 and 84-88).

  1. More detail concerning the data collection process is needed, namely to clarify the recruitment process, and eligibility criteria (never pregnant women in the control group?) ethic procedures, participation rate. Please describe in detail the setting, the procedures for data collection (who administered the questionnaire?), and the measures (indicate that weight and height were self-reported. How was BMI calculated for pregnant women? Why were only BMI and smoking selected to be independent variables?How was smoking status collected? Were there other socio-demographic characteristics besides age and ethnicity – which needs to be referred to in the methods section -, such as nationality and educational level collected?).

Response:

Recruitment of study participants and questionnaire administration was conducted by the author of the article.

No pregnant woman refused to participate in the study. In the control group, the percentage of participants was 97% and the refusal was mainly due to a reluctance to take a blood sample. There were no other socio-demographic characteristics besides age and ethnicity.

In accordance with the work procedures of the Israeli Ministry of Health, every new employee hired to work in a medical institution must be vaccinated against pertussis. The candidates are recruited in the occupational clinic, which is responsible for administering and documenting the vaccines for all employees in the medical centre. 

All participants received an explanation on the background and the purpose of the study, signed an informed consent form to participate in the study, provided their details and filled in the questionnaires. Institutional Review Board Statement: The institutional review board at Shamir (Asaf Harofe) Medical Center, Israel and ethics committee at Haifa University, Israel approved the study protocol, approval number 0290-17-ASF and 0181-17-ASF).

Medical records were checked to verify the pertussis vaccine administration. Pregnant and non-pregnant women who had previously received the pertussis containing vaccine or who got sick within the prior 5 years were excluded. In order to rule out vaccination in recent years only pregnant women with the first child participated in the study. In addition, every woman was asked about getting vaccinated for any reason. No booster doses were given 5 years before of serum sampling.

Pregnant women were asked about weight in pregnancy, so the BMI value of pregnant women is a derivative of their weight at the time of receiving the vaccine. Parameters as smoking status, height and weight were collected by self-report.

We have rewritten the paragraph (Methods, lines 84-88 and 91-105).

  1. Implications of the results for public policy should be enlightened. Future recommendations for research and action should be added.

Response:

The impact of sex hormones on the response to pertussis vaccine should be taken into account when evaluating the immune response to vaccination in pregnancy. Clinical and epidemiological studies should include the influence of the reproductive and hormonal status on vaccine-induced immunity. Non-pregnant and pregnant women should be considered as different groups when studying vaccine efficacy.

We have rewritten to make it clearer (lines 363-368)

  1. The English language (style and grammar) could be improved (some typos, consider replacing pregnant and non-pregnant female with pregnant and non-pregnant women).

We made the changes and replaced ''female'' with ''women''.

Round 2

Reviewer 2 Report

The authors have added some information on epidemiology of pertussis in the study population and explained the national vaccination strategy against pertussis toxin in more detail. This allows the reader to understand the epidemiological and immunological background of the study. Still, the authors could render the description and discussion of their results more convincing by taking the following suggestions into consideration:

  1. The authors should clarify AND unify the nomenclature regarding the immunological surrogate of response to vaccination they use: in a seemingly erratic manner they use at least the following descriptions as key words for antibodies against pertussis toxin: "pertussis" (Fig. 3), "IgG" (abstract), "PT-antibody", "PT-Ig-antibody", "anti-PT antibody". Please choose one of these key words, explain it when first mentioned, and use it throughout the text, figures, tables and abstract.
  2. In the discussion of findings (4.2) an explanation is still lacking how to explain the differences of findings in the authors' study and the papers of Munoz et al (ref. 15) and Huygen et al. (ref. 16). A paragraph could be added following line 280 of the manuscript.
  3. The discussion is still somehow difficult to read (and thus, not really convincing).  Avoiding some redundancy (line 291ff = line 314ff) it may be helpful to explain the general effects of pregnancy on the immune system first (e.g. by moving text from line 313ff. up to line 290ff.), followed by the explanation of specific aspects of anti-pertussis immune responses. Given the data they show, however, the authors cannot draw the conclusion that a high level of reproductive hormones has a suppressing effect on vaccine response (line 313f.). They may talk about correlation but not causality.

Regarding the "minor point" of "multiple comparisons" the authors' solution is somehow "cheap": aren't there rules how to deal with multiple comparisons that can be applied (e.g., adjusting the p-value that indicates statistical significance to the number of comparisons)?

Author Response

Reviewer 2

We thank you for the comprehensive review and suggestions.

  1. The authors should clarify AND unify the nomenclature regarding the immunological surrogate of response to vaccination they use: in a seemingly erratic manner they use at least the following descriptions as key words for antibodies against pertussis toxin: "pertussis" (Fig. 3), "IgG" (abstract), "PT-antibody", "PT-Ig-antibody", "anti-PT antibody". Please choose one of these key words, explain it when first mentioned, and use it throughout the text, figures, tables and abstract.

Response: The changes were made throughout the text. The chosen words are ''anti-PT IgG antibody''.

2.In the discussion of findings (4.2) an explanation is still lacking how to explain the differences of findings in the authors' study and the papers of Munoz et al (ref. 15) and Huygen et al. (ref. 16). A paragraph could be added following line 280 of the manuscript.

Response: In two studies [15, 16], there was no evidence of differences in the antibody responses to Tdap vaccine in pregnant and non-pregnant women. Pertussis IgG antibody levels increased significantly and to the same extent after vaccination in pregnant and non-pregnant women [15, 16].

In both studies [15, 16] the small number of participants were enrolled. Prior vaccination information about receiving the vaccine was obtained mainly through self-reporting which is a possible source of information bias [16]. 

It is possible that both studies were underpowered to detect any differences in the antibody responses between pregnant and non- pregnant females. (Please see '' Discussion '', lines 278-286).

 3. The discussion is still somehow difficult to read (and thus, not really convincing).  Avoiding some redundancy (line 291ff = line 314ff) it may be helpful to explain the general effects of pregnancy on the immune system first (e.g. by moving text from line 313ff. up to line 290ff.), followed by the explanation of specific aspects of anti-pertussis immune responses. Given the data they show, however, the authors cannot draw the conclusion that a high level of reproductive hormones has a suppressing effect on vaccine response (line 313f.). They may talk about correlation but not causality.

Response: The ''Discussion'' section was rewritten according to your suggestions.

  1. Regarding the "minor point" of "multiple comparisons" the authors' solution is somehow "cheap": aren't there rules how to deal with multiple comparisons that can be applied (e.g., adjusting the p-value that indicates statistical significance to the number of comparisons)?

Response: Indeed multiple comparisons and multiple hypothesis testing might yield significant findings by chance. However, in the current study we tested one main pre-defined hypothesis of an association between pregnancy and pertussis toxin IgG antibodies level. We also explored the levels of estrogen and progesterone in this association. Since biologically the levels of sex hormones differ substantially between pregnant and non-pregnant women, we planned stratified analyses separately for pregnant and non-pregnant women. The levels of estrogen and progesterone were highly correlated, therefore to avoid multicollinearity these variables were analyzed in separate models. Other variables such as age, body mass index, and smoking were included as potential confounders. Following the reviewer's comment' we added this information to the statistical methods section.  Beyond P values, the magnitude of the point estimate was generally strong for the main independent variables that were selected a-priori. Therefore, in our opinion the possibility of significant findings by chance should not be a main concern in our study. Epidemiologists have reservations regarding the need for correction for multiple comparisons (1-4). As mentioned, in our study we had one main hypothesis.  Nevertheless, we edited the methods section.

(Please see ''Methods'', lines 132-143).

Can the reviewer be more specific regarding the need for adjusting the multiple comparisons?

  1. Rothman KJ. No adjustments are needed for multiple comparisons. Epidemiology. 1990;1(1):43-6.
  2. Savitz DA, Olshan AF. Multiple comparisons and related issues in the interpretation of epidemiologic data. Am J Epidemiol. 1995;142(9):904-8.
  3. Savitz DA, Olshan AF. Describing data requires no adjustment for multiple comparisons: a reply from Savitz and Olshan. Am J Epidemiol. 1998;147(9):813-4; discussion 5.
  4. Perneger TV. What's wrong with Bonferroni adjustments. BMJ. 1998;316(7139):1236-8.

 5.Moderate English changes required

Response: We did linguistic editing of the text.